# Longitudinal Analysis of Sustainable Tourism Potential of the Black Sea Riparian States Bulgaria, Romania and Turkey

**DOI:** 10.3390/ijerph20042971

**Published:** 2023-02-08

**Authors:** Alina-Petronela Haller, Georgia-Daniela Tacu Hârșan

**Affiliations:** Romanian Academy, Branch of Iași—“Gheorghe Zane” Institute for Economic and Social Research, 700481 Iași, Romania

**Keywords:** tourism, sustainability, international tourism, Black Sea region, Bulgaria, Romania, Turkey

## Abstract

The three states that border the Black Sea benefit from an important potential for tourism and consider the development of this sector to be a major objective. Nonetheless, they face environmental risks. Tourism does not have a neutral impact on the ecosystem. We evaluated tourism sustainability for three states bordering the Black Sea, Bulgaria, Romania and Turkey. We used a longitudinal data analysis applied to five variables for the period between 2005 and 2020. The data were taken from the World Bank website. The results show that tourism receipts significantly influence the environment. For all three countries, the total receipts from international tourism are unsustainable, while the receipts for travel items are sustainable. Sustainability factors are different for each country. The international tourism expenditures for Bulgaria, the total receipts for Romania and the receipts for travel items for Turkey are sustainable. In Bulgaria, the receipts from international tourism contribute to higher greenhouse gas emissions, i.e., negative environmental impact. In Romania and Turkey, the number of arrivals has the same impact. No sustainable tourism model could be identified for the three countries. Tourism activity was found to be sustainable only due to the receipts for travel items, that is, indirectly, from tourism-related activities.

## 1. Introduction

Due to globalization, tourism is constantly developing at the worldwide level [1]. In Europe, for example, in the pre-pandemic period, tourism contributed greatly to economic growth and created jobs for millions of people, especially for youth, women and emigrants [2]. Tourism is, indeed, a key economic activity and an important source of income for European countries, with positive effects on growth, development and well-being [3].

The development of tourism was due to an improvement in living standards, as people had the chance to travel more often and more easily because transport became more affordable and destinations increased their touristic relevance by organizing more exciting events and providing more attractive tourist packages.

Tourism contributes to economic growth, stimulates the accumulation of physical, human, technological and innovative capital, promotes industries such as transport, hospitality and retailing, drives new revenue, encourages investments and infrastructure development, promotes competition on the labor market and disseminates knowledge [4]. It has a complex role and its effects are economic, social and psychological. It has a beneficial impact on poor areas as it revitalizes, modernizes and helps them develop [5]. A major effect is seen in the labor market, as tourism creates long-term jobs that do not require special qualifications or an academic background [6]. The literature recommends that poor countries take advantage of tourism specialization since it is a way of increasing income and consumption, and is, therefore, a way of development [7]. Tourism’s contribution to the labor market is all the more important as this sector absorbs labor force from vulnerable categories, i.e., young people, women and emigrants.

Tourism influences the local residents’ values and behavior, stimulates the economy, and impacts the environment [8]. As a result of the tourism sector expansion, visitor flows have increased, which has begun to affect the environment and biodiversity [9,10]. Authors have argued that tourism violates the principles of sustainable development because of its numerous negative environmental impacts [11]. Tourism economic benefits cannot be denied; however, the unsustainability risk is real; this is because in many destinations, tourism has turned into over-tourism [12]. Tourist visitation causes erosion and deterioration of heritage sites [13], and damages the environment and the ecosystem. If, economically, tourism provides advantages to local communities, ecologically, it is a challenge. Huge carbon dioxide emissions, a polluted environment and inhumane working conditions in tourist areas are but a few of the terrible consequences of over-tourism [11]. This is the reason why the concept of sustainable tourism development was created, i.e., to increase tourism activity without harming the environment and ensure the quality of life in the future [14]. Sustainable tourism is the only way to improve the tourism industry. It cannot be considered as a special form of tourism activity because all forms of tourism should become sustainable [15]. Sustainable tourism is expected to have a minimal negative impact on destinations and have as great a positive impact as possible, thanks to the contribution of local communities and administrations; tourism becoming a sustainable activity is, above all, their responsibility.

One of the most important issues facing modern tourism is mass tourism, and thus sustainable development and environmental concerns are now major aspects in planning this activity [16].

The development of any sector aggravates environmental degradation by increasing energy consumption and human activity in production, consumption, and transport. All of these are related to tourism and, therefore, their sustainability is a great concern wherever tourism is considered as a development objective. The three countries are no exception. People’s lifestyle has changed over the past few decades and they have a greater desire to travel. Tourists are more careful with the environment, abandon mass tourism and cope better with the idea of getting sick [17].

For the reasons above, we evaluated tourism sustainability for three countries bordering the Black Sea, Bulgaria, Romania and Turkey. Since economic analyses can extend from the smallest level to the global one and because there is no logical requirement that asks that a minimum number of entities be considered (analyses of regions of a country [18,19] can be as accurate as those of entire countries [20,21,22]), this study only deals with a three-entity group: the two Black Sea riparian states, Bulgaria and Romania, to which Turkey was added as a potential EU member state. In the following, we will review the literature, describe the methodology, discuss the results and draw conclusions.

## 2. Literature Review

The tourism literature is very rich and addresses complex aspects of sustainable tourism. Seasonal tourism destinations appear to be the least sustainable [23] (most studies even chose to ignore seasonality). Dogru et al. (2021) [24] argue that tourism development depends on factors related to economic growth, global and regional economic trends, and three aggregate indicators, namely the global growth effect, the global industrial mix effect and the global competitive share effect. Khan et al. (2019) [25] explored the link between greenhouse gas emissions and tourism, the index of financial development, energy consumption, renewable energy and trade for 34 developed countries in Asia, Europe and America over the period 1995–2017; the authors found an unidirectional causality from tourism to greenhouse gas emissions and renewable energy. Ben Jebli et al. (2019) [26] found long-term bidirectional causality between a series of variables, including tourism and greenhouse gas emissions.

Paramati et al. (2018) [27] concluded that tourism, mainly due to transport and accommodation, is responsible for high energy consumption, especially from fossil fuels, and the generation of large amounts of greenhouse gas emissions, which makes the tourism industry one of the most important sources of global warming. In Turkey and throughout the Black Sea region, tourism provides economic benefits to local communities, but they are unsustainable in the medium and long term [28].

There is a complex relationship between the economy, tourism, employment and the environment. Sustainability has a positive and statistically significant relationship with income (economic growth); it depends on the number of tourist arrivals (tourism), on gender-based employment and on the level of unemployment (labor market) [29].

Ehigiamusoe (2020) [30] argued that the impact of tourism on economic growth is not uniform across countries, as tourism and the economy do not have the same development level. This is because, in the first stage of development, tourism and the economy can only develop at the expense of the environment, mainly in the countries with no green technologies. As they develop, greenhouse gas emissions begin to decrease. The author argues that tourism’s environmental impact is strictly correlated with unsustainable tourism. Therefore, each country must decide to implement sustainable tourism through its development strategy.

Pérez-Rodríguez and Santana-Gallego (2020) [31] show that tourism revenue has important effects on destinations, that is, on budgetary revenue and the management of economic policies. Instead, Dogru et al. (2021) [24] argue that the changes in tourism receipts and tourist arrivals are an indicator of the tourism sector growth and economic competitiveness. Competitiveness ascribed to tourist arrivals means that tourists are interested in accessible products, while competitiveness ascribed to tourism receipts implies mass tourism, which leads to higher tourism expenditures and receipts, but also over-tourism and a negative environmental impact.

The impact of tourism can be measured by means of tourism receipts, but tourism expenditures [32,33] and the number of arrivals [4] are equally important. Analyzing how sustainable tourism helps reduce inequalities between European states, Haller (2021) [34] showed that international tourism expenditures, receipts and, to a lesser extent, tourist arrivals contribute to narrowing development gaps, emphasizing the importance of sustainable tourism.

Theorizing the concept of tourism specialization, Croes et al. (2021) [35] explained the importance of tourists’ expenses for accommodation, meals, sightseeing, purchases, transport, souvenirs and entertainment for economic growth. This money goes to companies as income, and they make investments; this includes investing in more eco-friendly activities, in people who purchase sustainable goods and services, and in public administration institutions through paying fees and taxes which, in turn, support sustainability measures. Tourism receipts generate sustainable growth and human development.

Receipts from international tourism have a positive and significant effect on public revenue [36]. The effect extends as the economy grows, hence the countries that benefit the most as a result of tourism expansion are the developed ones. Less developed countries are vulnerable to the changes in tourism expenditure from dominant economies [37]. Similar to the results of Usmani et al. (2020) [38], tourism expenditures have a positive impact on economic growth in all circumstances, while tourist arrivals have a much smaller impact.

Alola et al. (2021) [39] argue that tourist arrivals negatively impact the environment, and international tourism in general has the same impact in Turkey. Tourist arrivals affect, above all, residents’ satisfaction [40]; Sharma and Mitra (2021) [41] highlighted tourist arrivals’ impact on the labor market. The increase in the number of tourists stimulates job creation, both in tourism and the related sectors. This only happens in regions with an educated population because the educational advantage increases the opportunity to attract tourists [42]. Kilavuz et al. (2021) [43] examined the relationship between international tourist arrivals and carbon dioxide emissions per capita in Turkey over the period 1960–2015. As tourism demand increases, so does pollution, until the tourism sector reaches a certain level of development and the amount of greenhouse emissions begins to decrease.

Katircioğlu (2009) [44] did not find any cointegration between international tourism and economic growth in Turkey and rejected the hypothesis of tourism-led growth. In another study, the author [45] showed that tourism in Turkey had negative and statistically significant effects on CO_2_ emissions, in the short, as well as in the long term; this is as a result of a high energy consumption, which affects the environment and climate, even though it generates revenue growth. Sun et al. (2021) [46] concluded that tourism in Turkey contributed to higher environmental pollution, and Ozturk and Acaravci (2010) [47] explained that it was the result of increased energy consumption and carbon emissions.

Doncheva (2019) [48] concluded that tourism contributed to higher revenue in Bulgaria. Tourism’s economic impact was visible on the labor market, as jobs had been created. Therefore, attractive forms of tourism should be developed to reduce its seasonality. Another study disproving the negative environmental impact of tourism was conducted by Adedoyin et al. (2021) [49]. They argue that international tourism increases economic complexity, and that financial crises do not accelerate the environmental crisis in regions such as Romania and Bulgaria, where a one-unit increase in tourism activity causes an insignificant increase in carbon emissions; that is to say, tourism has no real impact on the environment.

The literature on Bulgaria, Romania and Turkey has placed little emphasis on the effects of tourism in terms of international tourism receipts, international tourism expenditures, receipts for travel items, and tourist arrivals. Likewise, there has been little emphasis on sustainability, evaluated in terms of the amount of greenhouse gas emissions. Golumbeanu et al. (2014) [50] maintain that pollution in the Black Sea area is the result of human activities, including over-tourism, which degrades the ecosystem in the region.

Due to the relevance of the tourism sector for the three countries, and the lack of studies on them, we will evaluate tourism sustainability in Bulgaria, Romania, and Turkey, starting with the following hypotheses:
**Hypothesis** **1 (H1).***Tourism in the Black Sea riparian region is sustainable, and its sustainability is due to international tourism expenditures, international tourism receipts, international tourism receipts for travel items, and international tourism number of arrivals*.
**Hypothesis** **2 (H2).***There exists a sustainable tourism model in the Black Sea riparian region.*
**Hypothesis** **3 (H3).***Tourism activity in the Black Sea riparian region is carried out under the conditions of sustainable economic growth*.

## 3. Materials and Methods

### 3.1. Data

We analyzed five variables, one for the environment and four for tourist activity, to which we added a growth variable. The indicators were taken from the World Bank database [46] for the time interval 2005–2020. This is the most compact time frame for which data was available. This time frame is the only one for which a complete database exists and is, therefore, the most suitable for the present study.

Similar studies using panel data covered, for example, a period of 12 years (2008–2019 [51], 2004–2016 [22]). The literature does not indicate that there is a minimum number of years for panel analyses to be valid [52,53]. In addition, Franses (2004) [54] explains how different the panels can be from one another. For example, in a condensed panel data set, either the number of individuals is equal to 1 or the data have been averaged over the *n* individuals, resulting in a single variable. As a rule, the more numerous the data in the panel, the more complex the model. A three-year panel is a simple one. Nevertheless, in the case of panel data analyses, the most relevant aspect is not the number of years, but the use of aggregated variables (instead of non-aggregated ones) [54].

Two criteria were used to select the indicators: intuition, based on the authors’ knowledge acquired over time on tourism sustainability; and evidence from similar studies. One indicator is the dependent variable, and the other four are the independent variables.

#### 3.1.1. The Dependent Variable

To describe environmental sustainability, we considered the total amount of greenhouse gas emissions per inhabitant (kt of CO_2_ equivalent); this is the total CO_2_ emissions, excluding short-cycle biomass burning, such as agricultural waste burning, but including other biomass burning, such as forest fires, and all anthropogenic sources of CH_4_, N_2_O, HFCs, PFCs, and SF_6_ [55]. Greenhouse emissions are dangerous because their chemical elements contribute the most to global warming [56].

The amount of greenhouse gas emissions is the dependent variable because this indicator describes the quality of the environment [18,20,51]. Campos et al. (2022) [19] made the same choice because the tourism sector had generated approximately 8% of total emissions.

Anthropogenic activity affects the environment through air pollution, water pollution, soil pollution, noise pollution, waste production, ecosystem damage, and biodiversity loss [57,58]. According to the UN [57,59], there are four ways to reduce the environmental impact: reduce the environmental footprint through climate neutrality, i.e., reduce the amount of greenhouse emissions and carbon dioxide emissions; integrate the climate change variable into business, i.e., comply with environmental, social and governmental criteria in terms of investments and define products with low emissions; promote a circular economy, i.e., minimize waste, change production and consumption behavior; and preserve biodiversity, i.e., protect the ecosystems that are more sensitive to the effects of climate change. Reducing greenhouse gas emissions is perhaps the most important way to protect the environment, as the others are derivative, and it is an indicator for which data are compact; this makes it the most studied aspect in environmental and sustainability analyses.

#### 3.1.2. The Independent Variables

To describe tourist activity, we considered international tourism expenditures (current USD per capita—ITE), international tourism receipts (current USD per capita—ITR), international tourism receipts for travel items (current USD per capita—ITRTI), and international tourism number of arrivals (number of tourists—ITNA). International tourism expenditures are expenditures made by outbound visitors in other countries, including their payments to foreign carriers for international passenger transport, together with expenditures by residents who make day trips abroad [55]. International tourism receipts are expenditures by international inbound visitors, including payments to national carriers for international transport [55]. International tourism receipts for travel items are the expenditures by international inbound visitors for purchased goods [55]. International tourism number of arrivals means the number of tourists who travel to a country for a period not exceeding one year, in order to carry out any activity that is not remunerated [55].

The tourism sector analysis was carried out on the following dimensions: attractiveness and satisfaction with the destination, economic dimensions, dimensions associated with the well-being of the local population and sustainability [60]. The variables in the paper cover these dimensions. The number of tourists visiting a certain destination shows its degree of attractiveness. The greater the number of tourists, the greater the attractiveness of the destination, and the great visitor satisfaction is. Tourism expenditures and receipts cover two dimensions, the economic dimension and that associated with the well-being of the local population. In order to survive in the market, tourism entrepreneurs need to convince visitors to spend money, as tourism receipts stimulate economic activity, job creation, investment, and support the economy and the well-being of residents. Therefore, the economic dimension means protecting the economic and living needs of the resident population, which is why it is assessed through gross domestic product, job creation and indicators that measure tourism performance (number of arrivals, overnight stays, length of stay, level of expenditure, housing occupancy rates, tourist company profits, prices and market shares) [61]. The tourism performance indicators that we have selected are available for the entire period under study and can be statistically evaluated with more accuracy. Tourism’s impact on the amount of greenhouse gas emissions corresponds to the dimension of sustainability. Tourism’s ecological footprint is assessed by its impact on climate change, especially as tourism in vulnerable areas can cause irreversible damage as a result of higher water and energy consumption, greenhouse gas emissions and biodiversity change [61].

The number of tourist arrivals was used as the independent variable in a study on the environmental impact of tourism in China [20]. Tourism expenditures and the resulting revenue support the development of the tourism sector, but at the same time contribute to environmental degradation. For example, food and drink vendors increase greenhouse gas emissions in the long run; all tourist activities, especially entertainment, gambling and recreation, contribute to a higher energy level; and economic growth has mixed impacts on greenhouse gas emissions and air pollution [21]. The number of tourist arrivals has negative effects on the well-being of local residents and the economic dimension, especially in the case of seasonal tourism [61]. International tourism expenditures, international tourism receipts and international tourism receipts for travel items describe the economic dimension of tourism. Several studies have focused on these variables as being representative of tourism.

The variables selected make it possible for the research hypotheses to be validated or invalidated. The first hypothesis, H1: *Tourism in the Black Sea riparian region is sustainable, and its sustainability is due to international tourism expenditures, international tourism receipts, international tourism receipts for travel items, international tourism number of arrivals*, is validated if the empirical analysis shows that the increase in the number of tourists in each of the three countries occurs in tandem with a reduction in emissions. In addition, if tourism expenditures and tourism receipts for travel items have a positive impact on the destination economy, their increase leads to a reduction in emissions. The second hypothesis, H2: *There exists a sustainable tourism model in the Black Sea riparian region*, is validated if the empirical analysis shows similar trends in the evolution of variables for the three countries. A sustainable tourism model in Bulgaria, Romania and Turkey implies that the increase in all the variables occurs simultaneously with the reduction in emissions. The third hypothesis, H3: *Tourism activity in the Black Sea riparian region is carried out under the conditions of sustainable economic growth*, requires an inductive and deductive analysis, in which the empirical results are theoretically correlated with the evolution of economic growth in Bulgaria, Romania and Turkey. The hypothesis is validated if we find, also from similar studies, that the reduction in greenhouse gas emissions occurs simultaneously with growing positive rates in the gross domestic product.

Compared to Bulgaria and Romania, Turkey dominates in terms of tourism. Between 2005 and 2020, more than 70% of greenhouse gas emissions came from Turkey. Turkey generates the largest contribution (37.81%) to the Black Sea region’s economic growth, followed by Romania (34.78%) and Bulgaria (27.41%). The largest contribution of tourism to the region’s exports comes from Turkey (50.79%), and the lowest from Bulgaria (17.8%). Turkey possesses a huge percentage of the total tourism receipts (82.04%), followed by Bulgaria (11.3%). Romania has the lowest percentage (6.66%). Turkey has the highest receipts for travel items (80.71%), and Romania the lowest. Turkey has the largest tourist flow, so it was perceived as the most attractive country in the region. More than 66% of visitors travelled to see Turkey. Romania and Bulgaria have a relatively similar degree of attractiveness. Bulgaria was more visited than Romania, but the regional gap is obvious. Turkey had a clear tourism superiority, but low sustainability. Turkey should increase its degree of sustainability, while Bulgaria and Romania should render their tourism sector more efficient and reduce regional gaps under sustainability conditions. Bulgaria and Romania need to increase their attractiveness. In addition, they need to raise tourists’ interest for their specific goods and services. Considering Turkey as a reference point for tourism development in the Black Sea riparian region, Bulgaria and Romania should adopt appropriate tourism development strategies, attractive tourism packages, and provide visitors with easy access to specific quality goods; all of this is under sustainability conditions.

### 3.2. Method

The panel data analysis we used refers to a set of indicators observed over time; this is the link for Bulgaria, Romania and Turkey between four independent variables: international tourism expenditures per capita (ITE), international tourism receipts (ITR), international tourism receipts for travel items (ITRTI) and international tourism number of arrivals (ITNA); and one dependent variable: the amount of greenhouse gas emissions per inhabitant (GHG).

The model was not randomly selected. Different combinations of variables and different variants of the panel model were tried. Finally, the most appropriate combination and the model with the most conclusive results were selected. The theoretical part of our empirical analysis was based on studies that employed the same empirical model [52,53,54,62].

After testing the fixed-effects and the random-effects regression models, we chose the first one. The fixed effects model was chosen following the application of three tests: the Hausman, Pesaran and modified Wald test.

In the case of the Hausman test, we compare two estimators, one coherent in both null and alternative hypotheses, and the other coherent only in the null hypothesis.

The Pesaran test is of the following form:(1)$$\mathrm{CD}=\sqrt{\frac{2T}{N\left(N-1\right)}}(\sum_{i=1}^{N-1}\sum_{j=i+1}^{N}\hat{pij}$$
where $\hat{pij}$ is sample estimate of the pairwise correlations of the residuals, *N* is the cross-sectional dimension and *T* is the panel’s time dimension.

The modified Wald test on a single parameter is of the following form:(2)$$W=\frac{({\hat{\theta -\theta_{0)}}}^{2}}{var(\hat{\theta )}}$$

The fixed effect model assumes that a characteristic of an entity may have an impact on the dependent variable, in our case, environmental sustainability. The model eliminates the effect of time-invariant characteristics, so that the effect of tourism-specific variables on sustainability can be estimated.
Y_it_ = β_i_ × X_it_ + α_i_ + μ_it_(3)

The general equation of the model is the one above (Equation (1)), where α_i_ (i = 1, …, n) is the unknown intercept for each individual entity, Y_it_ is the dependent variable, i is the entities, t is time, X_it_ is the independent variables, β_i_ is the coefficients of the independent variables, and μ_it_ is the error term.
u_it_ = μ_i_ + η_i_ + v_it_(4)

In Equation (2), u_it_ is the error term or discrepancy variable, μ_i_ is the unobservable, non-time-changing cross-sectional specific effect that estimates the effect of the variables not included in the model specific to the country and on the dependent variable, η_i_ is the temporary specific effect that does not change in the transversal structures, estimating the effect of the variables not included in the model, at time t, on the dependent variable, β_i_ is the coefficient of the independent variable and v_it_ is variable error among units (countries) and in space.

We assume that μ_i_ and η_i_ are fixed parameters, uncorrelated with errors, that the sum of them is zero, and the estimation method is that of the smallest squares.

Equation (1) can be written by inserting dummy variables in the form below (Equation (3)):(5)$$Y_{\mathrm{it}}=\sum_{j=1}^{N}\alpha_{j}+x_{\mathrm{it}}\beta +u_{\mathrm{it}}$$

The parameters *α_j_* and β can be estimated with ordinary least squares. The same estimation is obtained whether the regression is performed as a deviation from the individual mean *α_j_*, which involves removing individual effects by transforming the data [63]. The fixed effects term is justified by the fact that the intercept may vary between countries but not over time. The estimated coefficients of the fixed effects model cannot be influenced by time-invariant characteristics omitted from the analysis, such as culture, religion, gender, race.

## 4. Results and Discussion

Regarding tourism’s environmental impact in developing countries, there are four approaches, according to Gössling (2000) [64]. One is optimistic, characteristic of the 1960s–1970s, according to which the number of tourists increased significantly in these countries as a result of air transport development. Then, tourism was perceived as beneficial for development. This perception changed into criticism in the 1970s and 1980s when it was found that the economic benefits of tourism were negligible and that the sociocultural and environmental consequences were negative. In the 1980s and 1990s, the involvement approach was developed, which regarded tourism as an ambivalent phenomenon. On the one hand, the negative environmental impact became increasingly obvious, which, together with sociocultural concerns, increased the resistance from local residents. On the other hand, new concepts were developed, such as acceptable visitor density, maximum carrying capacity, ecotourism, and sustainable tourism. Along with the theorization of sustainable tourism, starting from the 1990s, the new optimism approach developed. Tourism, it was thought, had positive economic effects, creating income, jobs, infrastructure, reducing inequalities and developing rural areas. It also had an ecological impact, helping to conserve biodiversity and preserve heritage sites, and funding protected areas. Thus, through empirical analysis, we intend to find to what extent the new optimism approach is suitable for the three countries.

We applied the Levin–Lin–Chu test, whose null hypothesis (H_0_) means that the panel contains unit roots; the alternative (Ha) means that the panel is stationary (Table 1). The panel has no unit roots and is stationary, which allows the continuation of the empirical analysis (*p*-value = 0.0298) by selecting the appropriate variant of the estimated model using the Hausman test.

The model recommended after applying the Hausman test is the fixed effects model (Prob > chi2 = 0.0000). To be sure that it is appropriate, we applied the Pesaran test and the modified Wald test (Table 2). The results show that there is no serial correlation in the model because the probability value (0.2966) is above the significance threshold. The probability value makes us reject the alternative hypothesis and accept the null hypothesis of the test, namely, that there is no serial correlation; therefore, the fixed effect model is the right one. In the case of the modified Wald test, the residuals are homoscedastic (prob > chi2 = 0.2418). In the case of the modified Wald test, the probability value leads us to reject the alternative hypothesis of the test and to accept the null hypothesis, which assumes that the residuals are homoscedastic.

For the three countries, neither the international tourism expenditures (ITE) nor the number of arrivals (ITNA) are statistically significant variables (Table 3). International tourism receipts (ITR) and international tourism receipts for travel items (ITRTI) are statistically significant variables, the latter with a higher degree of significance than the former. Both of them have a significant influence on the amount of greenhouse gas emissions (GHG). The unstandardized coefficients fall within the confidence intervals and fit the model equation in the form below (Equation (6)).
GHG = 11.54 − 0.06 × ITE + 1.15 × ITR − 1.07 × ITRTI − 0.005 × ITNA(6)

If we ignore the influence of variables, the greenhouse emissions (GHG) in the Black Sea region increase (β_0_ = 11.54); thus, tourism could accelerate environmental degradation. If tourism has no environmental impact or even helps to reduce the emissions, it is sustainable. The values of unstandardized coefficients could imply that tourism reduces the emissions, since three of the four links are inverse (−0.06, −1.07, −0.005), and one is direct and significant (1.15). Therefore, a one-unit increase in ITE, ITRTI and ITNA leads to a reduction in emissions by 0.05 units, 1.06 units and 0.005 units, respectively. The international tourism number of arrivals (ITNA) has the lowest impact on the environment, but this variable is statistically insignificant. The international tourism receipts for travel items (ITRTI) have a positive impact; however, it cannot offset the impact of total tourism receipts that increase the amount of greenhouse gas emissions (GHG).

Each country has its own specificity (rho = 99.24%). Despite their geographical proximity and the development group they belong to, Bulgaria, Romania and Turkey are very different in terms of tourism’s impact on emissions. The proximity and the growth model could imply that the development of the tourism sector and its effects are similar in the three countries. Our results invalidate this hypothesis. Bulgaria, Romania and Turkey have dissimilar tourism development models and the influence on environmental sustainability is also different, according to the model configured by the tourism variables considered as a whole (87.42%), over time (56.8%), and simultaneously between countries and over time (41.95%).

The results partially validate the first hypothesis, H1: *Tourism in the Black Sea riparian region is sustainable, and its sustainability is due to international tourism expenditures, international tourism receipts, international tourism receipts for travel items, international tourism number of arrivals*, since only one component of the total tourism receipts contributes to the reduction in emissions through indirect effects, that is, the receipts for travel items intended, to some extent, to reduce the polluting effects of production.

The country-by-country analysis partly modifies the previous conclusions (Table 4).

None of the three countries follow the group model and there are dissimilarities between them, which partially validates the second hypothesis, H2: *There exists a sustainable tourism model in the Black Sea riparian region*. Without tourism, in Bulgaria and Turkey, the greenhouse emissions increase (β_0_ = 6.68. β_0_ = 6.99), which does not indicate a trend towards sustainability. Conversely, in Romania, the emissions decrease (β = −1.37), which is an indication of sustainability; however, this is due to factors other than tourism, especially the reduction in industrial activity.

For Bulgaria, the number of arrivals is not statistically significant (*p* > /t/= 0.116). Expenditures and receipts for travel items increase sustainability (β_ITE_ = −0.41; β_ITRTI_ = −0.66), while total tourism receipts do not provide a sustainable basis (β_ITR_ = 1.06).

For Romania, only the total receipts from tourism and the number of arrivals have statistical significance, but their environmental impact is not similar. Total tourism receipts reduce the level of emissions (β_ITR_ = −0.68), while the number of arrivals increases it (β_ITNA_ = 1.12).

For Turkey, the receipts for travel items and the number of arrivals are statistically significant variables. As for Bulgaria and Romania, they have different impact on emissions because the number of arrivals increases pollution (β_ITNA_ = 0.39).

For the group of countries, ITE and ITNA are statistically insignificant variables (hence unsustainable). For each country separately, ITE is statistically insignificant for Romania and Turkey, and ITNA is statistically insignificant for Bulgaria. ITE is sustainable for Romania and Bulgaria, but not for Turkey, and ITNA is unsustainable for Bulgaria and for the group. ITR, if it had not been statistically insignificant, would have contributed to higher emissions in Turkey, as for Bulgaria and for the group. ITR is sustainable in Romania only. The only sustainable variable both for each country and for the group is ITRTI, but it is not statistically significant for Romania.

These results partially validate the three hypotheses. The results for each country are relatively different from those for the group. None of the three countries have a common denominator with the group. Tourism has a sustainable component for the group and for each individual country, but each entity has its own specificity and the implications of tourism determinants are different from case to case. The determinants of regional sustainability do not coincide with those of each individual country. The common denominator is that tourism influences the amount of greenhouse gas emissions (GHG); therefore, it influences the sustainability of the three countries bordering the Black Sea.

The first hypothesis, H1: *Tourism in the Black Sea riparian region is sustainable, and its sustainability is due to international tourism expenditures, international tourism receipts, international tourism receipts for travel items, international tourism number of arrivals*, would have been validated if, for all three countries, the results had shown that the expansion of tourist activity occurred simultaneously with the reduction in emissions. This is only valid for Romania, although it is debatable.

The second hypothesis, H2: *There exists a sustainable tourism model in the Black Sea riparian region*, is not validated because the results do not indicate the existence of a sustainability model in the region. Each country has its own tourism model, with a different environmental impact.

The temporal dynamics analysis shows the same differences between the three countries. For Romania, the increase in international tourism receipts is sustainable (Table 5). In the case of ITR, there is no statistical significance for 2005, 2006, 2007 and 2013, and there is very little statistical significance for 2012, 2018, 2019 and 2020.

We could not identify a model of sustainable tourism for the three countries, even though international tourism receipts for travel items are sustainable. Receipts from international tourism do not reduce the greenhouse emissions in the region. International tourism expenditures have no statistical significance for the group, but they are a sustainability factor in Bulgaria. In Romania, the number of tourist arrivals has no statistical significance and total tourism receipts are sustainable. The factor unsustainable for the group is sustainable for Romania. For Turkey, tourism expenditures and receipts have no statistical significance, the receipts for travel items are sustainable, and the number of arrivals (with no statistical significance for the group and for Bulgaria) has negative effects (like in Romania) because it increases the greenhouse emissions.

The results of the temporal analysis partially validate the hypotheses and are similar to the previous ones (Table 6). Therefore, the first two hypotheses, H1: *Tourism in the Black Sea riparian region is sustainable, and its sustainability is due to international tourism expenditures, international tourism receipts, international tourism receipts for travel items, international tourism number of arrivals*, and H2: *There exists a sustainable tourism model in the Black Sea riparian region*, are validated for short and different periods of time for each of the three countries.

The group of the three countries is not characterized by sustainability (β_0_ = 11.54). Without tourist activity, only Romania has a trend towards sustainability (β_0_ = −1.37) due to the reduction in industrial economic activity, which results in the reduction in greenhouse gas emissions. Receipts for travel items are sustainable because their increase reduces the emissions. The goods purchased by tourists come from a variety of sources and the revenue they generate is used to implement green production technologies and models. Tourism expenditures are sustainable in Bulgaria and international tourism receipts are sustainable in Romania. Tourism activity is only sustainable due to the receipts for travel items, so indirectly, from tourism-related activities (Table 6).

For all three countries, economic growth is fluctuating, with a general upward trend. The emissions decreased constantly in Romania and Bulgaria, but not in Turkey. Bulgaria started to reduce emissions in 2012, and Romania, in 2017. The emissions (GHG) are still high in Turkey, which means that growth is less sustainable compared to the other two countries. Romania experiences a contraction of the GDP simultaneously with the decrease in emissions, while in Bulgaria, the GDP increases and the emissions decrease.

Not all the determinants of economic growth could be identified, which is a limitation of this study. The type of economic growth depends on the cumulative impact of all the determinants. Turkey’s economic growth is based on energy-consuming processes. Bulgaria made progress in applying an appropriate strategy for sustainable growth. Romania reduced its economic activity, and its sustainability is rather the result of the reduction in industrial activity, which is unfortunate for the Romanian economy and society. The average amount of emissions was highest in Turkey and lowest in Bulgaria. The average values of tourism indicators were higher in Turkey compared to Bulgaria and Romania, with a significantly higher average economic growth and a higher volume of emissions with environmental impact (Figure 1, Figure 2 and Figure 3).

In Bulgaria, ITE increased until 2010 and the upward trend was resumed in 2017; ITR increased slowly but steadily until 2020; and ITRTI decreased in 2010 compared to 2008, when the highest level was reached. The degree of tourist attractiveness is high as long as the tourist flow continues to rise. Bulgaria’s sustainable growth occurred amid a slow-growing tourism sector. Bulgaria made significant progress in terms of tourism sector development, capitalizing on the potential of the Black Sea, following the example of Greece and Turkey [65]. Tourism plays a key role in the Bulgarian economy and is able to stimulate the local economy. Bulgaria’s tourist assets are its natural diversity, history and its relatively small size [66]. After the 2008 crisis, the Bulgarian tourism sector was an engine of economic recovery [48]. It is a safe, affordable destination, with one of the fastest growing economies in the EU since the 2000s. The expansion of tourism in a relatively small country such as Bulgaria affected the environment and its natural resources, especially as a result of the development of infrastructure and additional amenities. The number of tourists visiting Bulgaria increased every year, which generated the construction sector expansion, especially hotel infrastructure and accommodation, mainly on the Black Sea coast [11]. A real challenge for Bulgaria is to increase competitiveness and economic performance while respecting the principles of sustainability, since, in the near future, tourism competitiveness will depend entirely on sustainability [48]. Bulgaria is committed to implementing the principles of sustainable development and tourism; it can contribute, directly and indirectly, to achieving this goal. In terms of biodiversity, the number of natural parks, reserves declared a biosphere by UNESCO, and heritage sites, some included in the List of World Heritage Sites, Bulgaria ranks second in Europe [11]. Bulgaria has many anthropogenic tourism resources that can prove valuable and effective through proper management, contributing to the country’s attractiveness [67]. The development of the tourism sector, the limitation of its negative effects and the development of economic growth opportunities, all led to the design of a sustainable development strategy in Bulgaria in 2014, updated in 2017, to be implemented by 2030 [68]. The strategy aims to achieve the following: implement green technology in transport; develop public transport and encourage tourists to use it; encourage tourists to spend more time in a destination in order to reduce the number of trips; increase the consumption of renewable energy in transport, thus combining sustainable transport with sustainable tourism; avoid waste, especially of water and energy; good waste management; conservation of biodiversity in tourist destinations; and implement landscaping for valuable urban structures and cultural resources [11]. The strategy aims to turn Bulgaria into a year-round destination, based on the principles of sustainability and with a focus on the following: the optimal use of resources; respect for the sociocultural authenticity of local communities; reliable economic activity and the optimal level of distribution of socio-economic benefits, which makes it necessary to involve authorities through pro-ecological policies that support the needs of present and future generations [48]. One of the sustainable tourism development measures is the development of the green skills of tourism employees. They do not require major investment since they refer to the use of low-energy technology and the application of measures such as saving energy and water by tourism companies; this also includes transferring the responsibility of saving energy and water to tourists [69]. Although Bulgaria has made progress, there is still much to achieve before its tourism sector becomes sustainable. Therefore, education should become part of tourism development strategies [69], not only in Bulgaria, but in all countries aiming to achieve sustainable tourism.

The Romanian tourism sector has developed considerably, but has not yet managed to reach a satisfactory level of competitiveness [70]. ITE fluctuated significantly, with a generally upward trend. Amid the 2008 financial crisis and possibly because of a less inspired tourism strategy than in Bulgaria and Turkey, ITR decreased and the same happened in 2016. ITR has grown slowly and we estimate that the values are still lower than in Bulgaria and Turkey, with reference to expenditures, different types of receipts and tourist flows, based on the evolution of these variables over the period 2005–2020 (Figure 1, Figure 2 and Figure 3). Most tourists who visit Romania are interested primarily in nature and secondly in culture. Its natural wealth is an excellent chance to turn Romania into a national and international ecotourism destination. Ecotourism evolved from isolated programs of local tour operators to integrated programs developed through partnerships between local authorities, communities and entrepreneurs [71]. Romania adopted a tourism development master plan in 2007, to be completed in 2026, but there is no indication of whether the tourism sector development policy is to be implemented or just configured in the coming years [68]. By 2030, Romania is supposed to become a high-quality and well-known tourist destination available year-round, that provides services at international standards and capitalizes on the uniqueness of its cultural and natural heritage [72]. In the long run, Romania aims to diversify and specialize its tourist offerings, and improve the quality of tourist services through the following: infrastructure investments; rehabilitation of degraded tourist areas; adequate promotion at national and international level, especially of rural, ecological and cultural tourism; preservation of traditional activities; preservation of traditional gastronomy and culture; and training and qualification of personnel [72]. However, there is no mention of developing green skills for tourism employees and entrepreneurs to ensure sustainability. Romania’s tourism development efforts are often hampered by blockages generated by the authorities and problems related to sustainability criteria, because the country has limited possibilities for economic diversification and income growth [73]. It seems that Romania has failed to create a competitive tourist product and capitalize on its natural potential. Added to these shortcomings are pollution and environmental degradation, to which tourism contributes through resource consumption, in the absence of a sustainable development framework supported by economic policy measures [74]. Tourism performance only exists in the big cities, on the Black Sea coast, in the Danube Delta and the Carpathians. Poor promotion led to the insufficient development of the tourism sector and to deficient legislation, especially for ecotourism, where sustainability depends on human potential, protected areas, the level of tourism, the degree of occupancy of accommodation spaces, population employed in tourism, the quality of accommodation spaces (drinking water and gas supply networks), road accessibility, and the quality of transport infrastructure [75]. All of these shortcomings have placed Romania in an inferior position in comparison to Turkey and Bulgaria in terms of tourism development.

Compared to the other two countries, Turkey is the favorite destination of foreign tourists. Tourism has been a key factor in Turkey’s economic growth since the 1980s, when many resources were allocated to the development of this sector [76,77]. It has developed to such an extent that tourism revenues are the second source of foreign currency income in the Turkish economy [77]. Turkish tourist indicators have higher values than in Romania and Bulgaria. ITE grew slowly, and so did ITE. Tourist flows maintained their upward trend. The year 2016 was an exception, when tourist indicators reached the lowest level. At the time, the economic growth was weak and tourism was severely affected by terrorist attacks, domestic political unrest and the war in Syria [78]. The decrease in the number of tourists in 2017 caused a decrease in GDP by one percentage point [78]. Turkey subsequently recovered its tourism sector and became one of the most popular tourist destinations in the world. The rise in the hierarchy of tourist destinations was due to the application of an inspired strategy and the efforts to develop the air transport network, the infrastructure (including the hotel one), an effective marketing policy, and the fact that only 3% of foreign tourists needed a visa [78]. Turkey has a high amount of generated emissions and an especially high consumption of energy from conventional sources [79]. In terms of sustainability, the tourism sector has several weak points: low renewable energy potential; insufficient conservation of natural heritage; poor sustainable environmental management; and insufficient recycling of solid waste [80]. Tourism, economic growth and energy consumption have a positive impact on the amount of emissions in the short and long run, and tourists are interested in the quality of the environment in the destination country [55]. All of this has strengthened the importance of sustainability in tourism more than in other sectors, and Turkey is concerned with developing sustainable tourism [77]. Turkey focuses on climate change [81] and is interested in environmental protection and the use of renewable energy [43]. In the Black Sea region, Turkey intends to focus on the development of rural tourism [82] and ecotourism, and also on strategic initiatives for sustainable tourism, that is, planning, managing and monitoring the tourism sector through long-term sustainable approaches [83]. Turkey makes effective use of its tourism industry and, together with Bulgaria, is one of the developing countries with a stable tourism industry [6]. Romania has a unique potential to attract tourists. However, it is below the level of Turkey and Bulgaria because of a less efficient tourism industry due to a poorly developed infrastructure and a low degree of international openness. By correlating tourism indicators with economic growth, we found that tourism influenced growth in all three countries; however, in terms of sustainability, this was more in Bulgaria and Romania than in Turkey, and more when tourism was included in a broad development strategy.

The year 2020 was a crossroads. In recent decades, the world has faced several pandemics, but none had the implications of the SARS-CoV-2 infection [84]. Pandemic restrictions significantly reduced the number of tourists, especially in highly exposed countries and those that took drastic measures. At the same time, restrictions caused a decrease in greenhouse gas emissions from tourism in all CEE countries, with different scales of impact from country to country; meanwhile, the relationship between tourism development and pollution levels was positive [85]. The pandemic exposed the weaknesses of the tourism sector and its inability to manage large-scale crises. The restrictions reduced travel and the entire tourism infrastructure. The infrastructure was the most affected as it is highly vulnerable to events with a risk to personal safety, security, and health [86]. None of the CEE countries experienced the impact of the crisis caused by the pandemic in the same way that Western European countries experienced it [70]. Bulgaria, Romania and Turkey did not impose strict measures but, nevertheless, the impact of restrictions on tourism and that of tourism on the environment should be approached with caution [84].

Pandemic restrictions affected Bulgarian tourism. However, people were optimistic and willing to return to their travel habits, more inclined to travel within the country, with family, and in hygienic and safe conditions [86].

In Romania, the restrictions blocked tourist activity in the first part of 2020 [87]. The small firms were the most affected, without liquidity and with difficulty adapting.

In the second half of 2020, tourism activity began to recover mainly due to the development of domestic, family and individual tourism, in less crowded areas. The entire tourism sector needs support from all public institutions. Unfortunately, in the Romanian tourism industry there were no coherent measures, the restrictions were relaxed too slowly, and the supply did not keep up with the demand, which prevented the timely entry of tourist packages adapted to the new reality generated by the pandemic to the market [88]. In Turkey, the tourism sector was recovering and adapting to changes in tourism demand [89], which proves that it is relatively resilient. The post-pandemic reactions of the three countries show that Turkey and Bulgaria are able to adapt relatively easily to new realities. Romania has an inertial, disadvantageous behavior, even though in terms of sustainability it has some advantage; this is not, however, due to sustainable tourism.

The conclusions above, drawn from our empirical analysis and the literature, partially validate the third hypothesis, H3: *Tourism activity in the Black Sea riparian region is carried out under the conditions of sustainable economic growth*. Economic growth reaches a different level in each of the three countries. In addition, general economic growth and tourism-based economic growth have different effects on environmental sustainability.

## 5. Conclusions

Tourism is blamed for not being able to support sustainable development because of its environmental impact, that is, it generates a large amount of greenhouse gas emissions through over-tourism and transport. In this context, we evaluated the sustainability of tourism in Bulgaria, Romania, and Turkey. We analyzed sustainability by means of greenhouse gas emissions, and tourism by four variables with direct and indirect impact: international tourism expenditures (ITE), international tourism receipts (ITR), international tourism receipts for travel items (ITRTI), and international tourism number of arrivals (ITNA). We formulated three hypotheses, which were partially validated.

Tourism is relatively sustainable for the group of the three countries and for each individual country. The explanatory variables were, in some cases, statistically significant and in others, they were not. Even though their geographical position and proximity, especially for Romania and Bulgaria, would indicate the existence of a model, this assumption proved false. For the region as a whole, ITE and ITNA are not statistically significant, and ITR have no sustainable impact. ITE and ITRTI are sustainable for Bulgaria, ITR for Romania, and ITRTI for Turkey. Tourist flows are not statistically significant and, therefore, we cannot estimate their impact on sustainability. If we ignore the influence of the tourism sector, in Bulgaria and Turkey, the amount of greenhouse gas emissions increases. Turkey seems to have an unsustainable, energy-intensive growth, and tourism is no exception. Bulgaria is striving to achieve sustainable growth. Romania has managed to apply effective sustainability measures, although the reduction in emissions may be due to the contraction of economic activity in general and industrial activity in particular. However, the statement, to be proven, requires a more complex analysis. Certain aspects of tourist activity determine, in one way or another, the amount of greenhouse gas emissions.

The development of Turkey’s tourism sector is superior compared to Bulgaria and Romania. Turkey attracts the largest number of tourists, earns the largest total revenue from international tourism and travel items, and its tourism contributes the most to exports, being an important source of foreign currency. However, this performance is achieved in rather unsustainable conditions. Turkey’s main problem is sustainability. The main problem of Bulgaria and Romania is the reduction in tourism gaps in terms of sustainability. For the group of the three countries, the receipts for travel items are invested in green technology and efficient methods to reduce greenhouse gas emissions. However, this does not happen in each of the three countries because there is no sustainable tourism model.

The conclusion of this study is twofold: (1) the tourism sector as a whole is not sustainable in Bulgaria, Romania and Turkey; and (2) some of its characteristics are sustainable. The new optimism approach [64] fits the reality of the three countries, because the complexity of the link between economy and environment makes it difficult to achieve sustainability goals; however, they are also in line with the implication approach developed in the 1980s and 1990s [64], which stipulates that tourism has ambiguous effects.

Some authors conclude that the tourism sector has a harmful effect on the environment [25,26,27,28,48], while others argue that tourism does not have a negative environmental impact [44,46,49]. Our conclusion is that tourism has a relatively sustainable dimension.

Our results indicate the existence of major deficiencies in the Black Sea riparian region, both in terms of the development of the tourism sector and the environmental impact. Bulgaria, and especially Romania, despite their high tourism potential, need to strive to improve their tourist offerings and attract enough tourists to become strong competitors on the European tourism market. All three countries need to focus on sustainability and eco-friendly tourism activities. They need appropriate conjunctural and structural economic policies and entrepreneurial policies because changing production and consumption behavior is the way to achieve the goals of sustainable development and competitiveness. Our results demonstrate the lack of tourism convergence in the Black Sea region, despite the EU objective to bridge the gaps between members states. In addition, the pressure to reduce the carbon footprint has a different impact in each of the three countries. The tourism sector growth is rather unsustainable in Bulgaria and Turkey. In Romania, competitiveness is constantly decreasing in regional tourism, although progress toward sustainability has been made, but for rather questionable reasons.

The limitations of this study are due to the variables analyzed and the methodology used. Undoubtedly, the number of variables, their impact, the time frame, and the empirical method influence the results of any research. The analysis of other tourism variables could change the results or bring clarifications regarding the sustainable nature of tourism activity. Another limitation is that, apart from Bulgaria, Romania and Turkey, no other country bordering the Black Sea was considered. As such, the number of the riparian states can be expanded in the future.

The results of this study can contribute to the elaboration of tourism development strategies aimed at reducing tourism environmental impact through a better understanding of the relationship between tourism and the environment for each of the three countries. This study opens up future research opportunities: more variables, a larger number of countries, a longer time frame, and even a new methodology could be considered.

## Figures and Tables

**Figure 1 ijerph-20-02971-f001:**
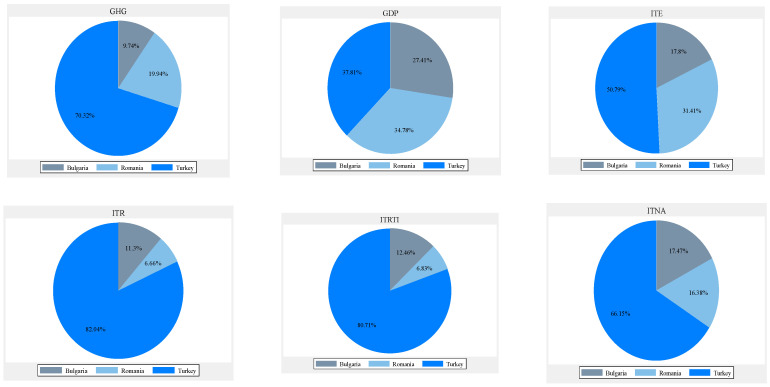
Average values of variables (2005–2020).

**Figure 2 ijerph-20-02971-f002:**
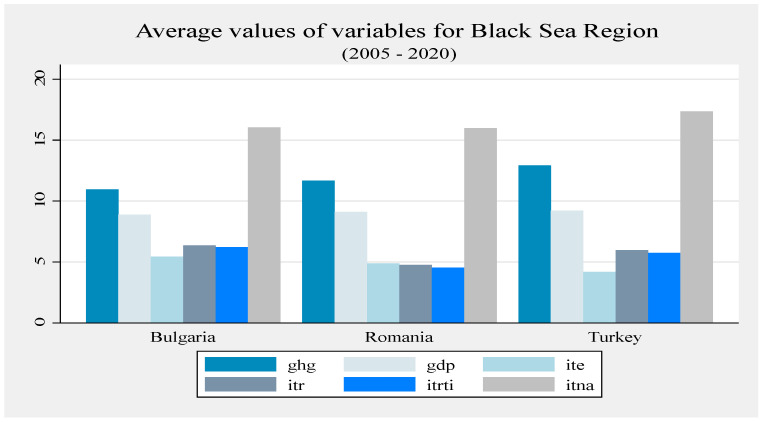
Average values of variables for Black Sea region (2005–2020).

**Figure 3 ijerph-20-02971-f003:**
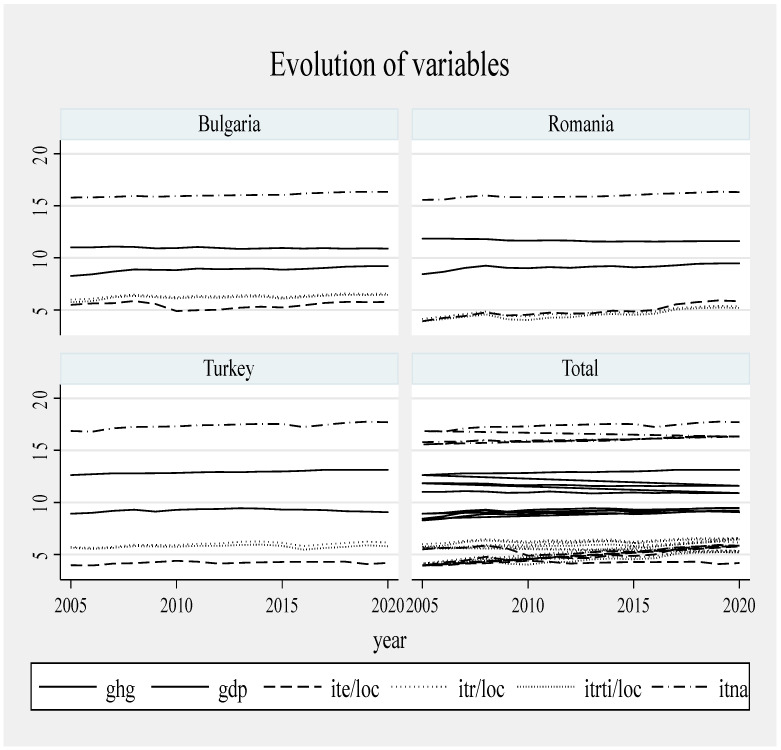
Evolution of variables. Source: [46].

**Table 1 ijerph-20-02971-t001:** Levin-Lin-Chu Test.

	Statistic	*p*-Value
Unadjusted t	−2.5729	0.0298
Adjusted t	−1.8844	

**Table 2 ijerph-20-02971-t002:** Hausman Test.

Variables	Coefficients		Chi2(4)	Prob > Chi2
	Fixed	Random		
ITE	−0.0578	−0.3203	123.07	0.0000
ITR	1.1451	−0.2859		
ITRTI	−1.0650	−0.0929		
ITNA	−0.0052	1.0406		
Pesaran’s test	1.379		Pr = 0.2966	
Modified Wald test	0.21		0.2418	

**Table 3 ijerph-20-02971-t003:** Tourism impact on sustainability for the three countries taken as a group (2005–2020).

Variable	Coefficients	t	*p* > /t/	95% Confidence Intervals	
ITE	−0.0578	−1.12	0.271	−0.1624	0.0468
ITR	1.1451	4.22	0.000	0.5968	1.6934
ITRTI	−1.0650	−5.62	0.000	−1.4475	−0.6826
ITNA	−0.0052	−0.04	0.967	−0.2556	0.2452
_cons	11.5429	7.86	0.000	8.5752	14.5106
rho	0.9925				
R-squared:					
within between overall	0.5680 0.8742 0.4195				
F(4, 41) Prob > F	13.48 0.0000				

**Table 4 ijerph-20-02971-t004:** Tourism impact on sustainability in Bulgaria, Romania and Turkey (2005–2020).

Country	Variable	Coefficient	t	*p* > /t/
Bulgaria	ITE	−0.4060	−7.87	0.000
	ITR	1.0564	2.39	0.000
	ITRTI	−0.6611	−2.20	0.033
	ITNA	0.3106	1.6	0.116
	_cons	6.6814	2.94	0.005
Romania	ITE	−0.1158	−1.28	0.209
	ITR	−0.6856	−2.04	0.047
	ITRTI	−0.1090	−0.39	0.701
	ITNA	1.1227	18.67	0.000
	_cons	−1.3775	−1.59	0.120
Turkey	ITE	0.0360	0.67	0.506
	ITR	0.3461	1.63	0.111
	ITRTI	−0.7712	−3.8	0.000
	ITNA	0.3969	4.75	0.000
	_cons	6.9908	6.00	0.000

**Table 5 ijerph-20-02971-t005:** Tourism impact on sustainability in Bulgaria, Romania and Turkey (temporal dynamics).

	Bulgaria			Romania			Turkey		
	coef	t	*p* > /t/	coef	t	*p* > /t/	coef	t	*p* > /t/
	2005								
ITE	−0.38	−7.4	0.000	−0.12	−1.3	0.201	0.036	0.68	0.501
ITR	1.16	2.6	0.011	−0.65	−1.87	0.068	0.35	1.62	0.113
ITRTI	−0.7	−2.5	0.017	−0.12	−0.42	0.675	−0.77	−3.77	0.001
ITNA	0.3	1.6	0.120	1.12	18.48	0.000	0.38	4.25	0.000
cons	6.6	2.97	0.005	−1.51	−1.61	0.114	7.22	5.66	0.000
	2006								
ITE	−0.4	−7.44	0.000	−0.14	−1.56	0.127	0.03	0.52	0.605
ITR	0.97	2.33	0.025	−0.57	−1.76	0.085	0.33	1.52	0.137
ITRTI	−0.65	−2.3	0.027	−0.13	−0.48	0.634	−0.75	−3.65	0.001
ITNA	0.39	2.13	0.039	1.13	19.45	0.000	0.42	4.55	0.000
cons	5.51	2.53	0.015	−1.91	−2.19	0.034	6.53	4.93	0.000
	2007								
ITE	−0.4	−7.9	0.000	−0.12	−1.34	0.186	0.03	0.65	0.521
ITR	1.13	2.65	0.011	−0.58	−1.74	0.090	0.39	1.89	0.066
ITRTI	−0.74	−2.57	0.014	−0.19	−0.68	0.502	−0.81	−4.15	0.000
ITNA	0.33	1.77	0.085	1.12	19.21	0.000	0.42	5.21	0.000
cons	6.37	2.93	0.006	−1.58	−1.86	0.070	6.57	5.79	0.000
	2008								
ITE	−0.4	−7.77	0.000	−0.11	−1.26	0.216	0.03	0.51	0.611
ITR	0.99	2.22	0.032	−0.66	−2.03	0.048	0.35	1.73	0.092
ITRTI	−0.64	−2.16	0.037	−0.15	−0.54	0.589	−0.78	−3.97	0.000
ITNA	0.36	1.83	0.074	1.12	19.42	0.000	0.41	5.09	0.000
cons	6.1	2.64	0.01	−1.39	−1.66	0.104	6.76	5.96	0.000
	2009								
ITE	−0.41	−7.76	0.000	−0.11	−1.24	0.222	0.04	0.72	0.479
ITR	1.07	2.37	0.022	−0.69	−2.03	0.049	0.35	1.62	0.113
ITRTI	−0.67	−2.2	0.033	−0.11	−0.37	0.712	−0.77	−3.79	0.000
ITNA	0.31	1.58	1.123	1.12	18.45	0.000	0.39	4.57	0.000
cons	6.69	2.91	0.006	−1.36	−1.54	0.130	7.14	6.02	0.000
	2010								
ITE	−0.41	−7.82	0.000	−0.11	−1.27	0.212	0.03	0.55	0.584
ITR	1.03	2.29	0.027	−0.7	−2.1	0.042	0.35	1.70	0.097
ITRTI	−0.65	−2.15	0.038	−0.12	−0.42	0.680	−0.78	−3.90	0.000
ITNA	0.32	1.64	0.109	1.12	18.77	0.000	0.39	4.75	0.000
cons	6.57	2.85	0.001	−1.2	−1.39	0.173	7.13	6.22	0.0000
	2011								
ITE	−0.41	−7.76	0.000	−0.11	−1.18	0.244	0.04	0.7	0.490
ITR	1.08	2.39	0.021	−0.7	−2.06	0.046	0.34	1.58	0.122
ITRTI	−0.68	−2.22	0.032	−0.1	−0.35	0.727	−0.77	−3.73	0.001
ITNA	0.3	1.5	0.142	1.13	18.29	0.000	0.39	4.71	0.000
cons	6.85	2.94	0.005	−1.42	−1.61	0.116	6.97	5.92	0.000
	2012								
ITE	−0.41	−7.92	0.000	0.13	−1.39	0.172	0.03	0.58	0.563
ITR	1.04	2.33	0.025	−0.67	−1.99	0.053	0.34	1.59	0.120
ITRTI	−0.65	−2.16	0.036	−0.11	−0.37	0.710	−0.77	−3.73	0.001
ITNA	0.32	1.63	0.111	1.12	18.29	0.000	0.39	4.7	0.000
cons	−1.09	2.93	−0.006	−1.3	−1.48	0.147	6.96	5.89	0.000
	2013								
ITE	−0.44	−8.96	0.000	−0.13	−1.35	0.194	0.03	0.52	0.607
ITR	1.43	3.31	0.002	−0.65	−1.88	0.067	0.36	1.67	0.103
ITRTI	−0.85	−2.97	0.005	−0.12	−0.41	0.683	−0.78	−3.82	0.000
ITNA	0.13	0.71	0.484	1.12	17.72	0.000	0.4	4.78	0.000
cons	8.76	3.94	0.000	−1.35	−1.53	0.133	6.94	5.94	0.000
	2014								
ITE	−0.42	−8.15	0.000	−0.11	−1.22	0.231	0.04	0.65	0.519
ITR	1.2	2.7	0.010	−0.69	−2.01	0.051	0.35	1.61	0.115
ITRTI	−0.73	−2.45	0.019	−0.11	−0.38	0.707	−0.77	−3.76	0.001
ITNA	0.24	1.22	0.229	1.23	18.24	0.000	0.4	4.69	0.000
cons	7.5	3.27	0.002	−1.38	−1.57	0.124	6.98	5.91	0.000
	2015								
ITE	−0.4	−7.7	0.000	−0.11	−1.28	0.208	0.03	0.56	0.577
ITR	0.99	2.18	0.035	−0.7	−2.13	0.040	0.33	1.54	0.131
ITRTI	−0.63	−2.07	0.045	−0.09	−0.34	0.735	−0.76	−3.69	0.001
ITNA	0.34	1.69	0.098	1.13	18.97	0.000	0.4	4.8	0.000
cons	6.34	2.7	0.010	−1.4	−1.64	0.109	6.84	5.78	0.000
	2016								
ITE	−0.41	−7.65	0.000	−0.09	−1.06	0.297	0.04	0.74	0.461
ITR	1.05	2.31	0.026	−0.69	−2.09	0.043	0.36	1.7	0.097
ITRTI	−0.66	−2.17	0.036	−0.13	−0.46	0.644	−0.78	−3.9	0.000
ITNA	0.31	1.57	0.125	1.13	18.82	0.000	0.39	4.82	0.000
cons	6.63	2.81	0.007	−1.43	−1.66	0.104	6.94	6.01	0.000
	2017								
ITE	−0.41	−7.73	0.000	−0.12	−1.3	0.202	0.03	0.57	0.570
ITR	1.06	2.36	0.023	−0.68	−2.02	0.049	0.34	1.61	0.115
ITRTI	−0.66	−2.18	0.035	−0.11	−0.37	0.713	−0.77	−3.76	0.001
ITNA	0.30	1.54	0.130	1.12	18.18	0.000	0.39	4.64	0.000
cons	6.81	2.95	0.005	−1.31	−1.47	0.150	7.08	8.4	0.000
	2018								
ITE	−0.4	−7.32	0.000	−0.11	−1.2	0.238	0.04	0.64	0.523
ITR	1.05	2.34	0.024	−0.68	−2.0	0.052	0.35	1.61	0.116
ITRTI	−0.66	−2.18	0.035	−0.11	−0.4	0.690	−0.77	−3.75	0.001
ITNA	0.32	1.6	0.118	1.13	18.14	0.000	0.39	4.59	0.000
cons	6.59	2.83	0.007	−1.47	−1.6	0.118	7.01	5.79	0.000
	2019								
ITE	−0.39	−7.17	0.000	−0.11	−1.24	0.222	0.04	0.66	0.514
ITR	1.04	2.34	0.024	−0.66	−1.94	0.059	0.35	1.62	0.113
ITRTI	−0.67	−2.21	0.032	−0.12	−0.42	0.675	−0.77	−3.77	0.001
ITNA	0.34	1.72	0.094	1.12	18.43	0.000	0.38	4.36	0.000
cons	6.28	2.71	0.010	−0.54	−1.72	0.094	7.11	5.65	0.000
	2020								
ITE	−0.39	−7.24	0.000	−0.11	−1.21	0.232	0.03	0.64	0.526
ITR	1.05	2.33	0.025	−0.68	−1.99	0.054	0.35	1.62	0.113
ITRTI	−0.66	−2.19	0.035	−0.12	−0.4	0.689	−0.77	−3.76	0.001
ITNA	0.32	1.64	0.108	1.13	18.25	0.000	0.39	4.47	0.000
cons	6.47	2.77	0.008	−1.51	−1.63	0.110	7.09	5.75	0.000

**Table 6 ijerph-20-02971-t006:** Tourism impact on the amount of greenhouse gas emissions (tourism sustainability).

	Variables				Non-Standardized Coefficient
	ITE	ITR	ITRTI	ITNA	
Group	No	Yes (+)	Yes (-)	No	11.54
Bulgaria	Yes (-)	Yes (+)	Yes (-)	No	6.68
Romania	No	Yes (-)	No	Yes (+)	−1.37
Turkey	No	No	Yes (-)	Yes (+)	6.99

## Data Availability

The data used to generate the results of this study were taken from the World Bank database (https://data.worldbank.org/).

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
