# Peer review of "Longitudinal Analysis of Sustainable Tourism Potential of the Black Sea Riparian States Bulgaria, Romania and Turkey"

_ijerph, 2023, doi:10.3390/ijerph20042971_

Round 1

Reviewer 1 Report

The authors noticed an extremely interesting issue of the relationship between sustainable tourism and sustainable development of the country, and the risk of a lack of a sustainable approach is very high in this area.

The literature review on tourism and sustainable development is interesting and seems to be complete.

The selection of indicators, whose analysis is to help verify the adopted hypotheses, is not very clear. Why such indicators were chosen? Have them come from either a logical justification or from an indication of other studies that have taken the same indicators into account.

It is not very clear which variables are dependent and which variables are independent. Why are greenhouse gases used to determine the level of sustainable development, and why not other elements of sustainable development? The statistical analysis seems to be correct, but the choice of variables is not well explained.

Recommend supplementing the article with an indication of the reasons for the selection of indicators and an explanation of why the relationships between them are to be the factors for verifying the adopted hypotheses.

Author Response

Dear Sir / Madam,

We received your evaluation on the paper Longitudinal analysis of sustainable tourism potential of the Black Sea riparian states Bulgaria, Romania and Turkey. The suggestions you made helped us improve our work. Thank you for your time and effort!

Please find below the answers to your observations (colored in red, like in the paper).

The authors noticed an extremely interesting issue of the relationship between sustainable tourism and sustainable development of the country, and the risk of a lack of a sustainable approach is very high in this area.

The literature review on tourism and sustainable development is interesting and seems to be complete.

Thank you for your appreciations.

The selection of indicators, whose analysis is to help verify the adopted hypotheses, is not very clear. Why such indicators were chosen? Have them come from either a logical justification or from an indication of other studies that have taken the same indicators into account.

We added explanations in Section 3.1 (3.1.1. and 3.1.2), as follows:

â–ª Page 5

— Two criteria were used to select the indicators: intuition, based on authors’ knowledge acquired over time on tourism sustainability; evidence from similar studies. One indicator is the dependent variable, the other four are the independent ones.

— The amount of greenhouse gas emissions is the dependent variable because this indicator describes the quality of the environment [75, 76, 77]. Campos et al. (2022) [78] made the same choice because the tourism sector had generated approximately 8% of total emissions.

â–ª Page 6

— The number of tourist arrivals was used as independent variable in a study on the environmental impact of tourism in China [77]. Tourism expenditures and the resulting revenue support the tourism sector development, but at the same time contribute to environmental degradation. For example, food and drink places increase greenhouse gas emissions in the long run; all tourist activities, especially entertainment, gambling and recreation, contribute to higher energy level; economic growth has mixed impacts on greenhouse gas emissions and air pollution [79]. The number of tourist arrivals has negative effects on the well-being of local residents and the economic dimension, especially in the case of seasonal tourism [83]. International tourism expenditures, international tourism receipts and international tourism receipts for travel items describe the economic dimension of tourism. Several studies focused on these variables as representative for tourism.

It is not very clear which variables are dependent and which variables are independent. Why are greenhouse gases used to determine the level of sustainable development, and why not other elements of sustainable development? The statistical analysis seems to be correct, but the choice of variables is not well explained.

We divided Section 3.1 (Data) into two sub-sections, explaining why we chose the dependent variable (3.1.1) and the independent variables (3.1.2.).

Recommend supplementing the article with an indication of the reasons for the selection of indicators and an explanation of why the relationships between them are to be the factors for verifying the adopted hypotheses.

We explained as follows:

â–ª Page 6

— The variables selected make it possible for the research hypotheses to be validated or invalidated. The first hypothesis, H1: Tourism in the Black Sea riparian region is sustainable, and its sustainability is due to international tourism expenditures, international tourism receipts, international tourism receipts for travel items, international tourism number of arrivals, is validated if the empirical analysis shows that the increase in the number of tourists in each of the three countries occurs simultaneously with the reduction in emissions. Also, if tourism expenditures and tourism receipts for travel items have a positive impact on the destination economy, so that their increase leads to the reduction in emissions. The second hypothesis, H2: There exists a sustainable tourism model in the Black Sea riparian region, is validated if the empirical analysis shows similar trends in the evolution of variables for the three countries. A sustainable tourism model in Bulgaria, Romania and Turkey implies that the increase in all variables occurs simultaneously with the reduction in emissions. The third hypothesis, H3: Tourism activity in the Black Sea riparian region is carried out under the conditions of sustainable economic growth, requires inductive and deductive analysis in which the empirical results are theoretically correlated with the evolution of economic growth in Bulgaria, Romania and Turkey. The hypothesis is validated if we find, also from similar studies, that the reduction in greenhouse gas emissions occurs simultaneously with growing positive rates of the gross domestic product.

We hope that we have managed to answer your requirements. We will try to further improve the text if necessary.

Yours faithfully,

Authors

Reviewer 2 Report

This paper provides new empirical evidence on the relationship between CO2 emissions and tourism. A static fixed effects model is used. The data set is extremely small, T=15 and N=3 (Bulgaria, Romania and Turkey). Please extend the dataset (more years and more countries) use a dynamic panel data model (PMG or the correlated variants).

The amount of GHG emissions per inhabitant depends on many factors. Energy production and processing and transport of inhabitants (not tourism) are the largest sources. Please add other explanatory variables. The equation suffers from an "omitted variable bias".

The approach to modelling the impact of tourism on emissions is wrong. Please read carefully Lenzen et al. (2018) which is the seminal reference. 

Please elaborate on the contribution. There are so many empirical studies investigating the relationship between total co2 emissions and tourism using aggregate country level data. 

Reference

Lenzen, M., Sun, Y. Y., Faturay, F., Ting, Y. P., Geschke, A., & Malik, A. (2018). The carbon footprint of global tourism. Nature Climate Change, 8(6), 522-528.

Pesaran, M. H., Shin, Y., & Smith, R. P. (1999). Pooled mean group estimation of dynamic heterogeneous panels. Journal of the American statistical Association, 94(446), 621-634.

Author Response

Dear Sir / Madam,

We received your evaluation on the paper Longitudinal analysis of sustainable tourism potential of the Black Sea riparian states Bulgaria, Romania and Turkey. Thank you for your time and effort!

Please find below the answers to your observations (colored in blue, like in the paper).

This paper provides new empirical evidence on the relationship between CO2 emissions and tourism. A static fixed effects model is used. The data set is extremely small, T=15 and N=3 (Bulgaria, Romania and Turkey). Please extend the dataset (more years and more countries) use a dynamic panel data model (PMG or the correlated variants).

The analysis targets the Black Sea riparian region. The countries that can be included are Bulgaria, Georgia, Romania, Russia, Turkey, and Ukraine. We have included only 3 countries for several reasons. First of all, Romania and Bulgaria are EU members, they are neighboring countries and have approximately the same model and development trends. Turkey, a potential member of the EU, has made remarkable progress in tourism. Georgia, Russia and Ukraine are not within the scope of our scientific concerns because they are tourist markets less accessible to the European population.

The time interval is 15 years because we used secondary data. As known, secondary data is taken from databases to which one has access. In our case, the World Bank. Since databases depend on the statistically available information, we chose the independent variables based on data availability. Of all the indicators that the World Bank makes available for the 3 countries and that describe the performance of the tourism sector, we have created the most complete database (4 indicators covering the period 2005-2020). Extending the period involves the risk of relying on incomplete data and thus certain variables should be eliminated.

We added the following paragraph, for greater clarity of our intentions and results:

â–ª Page 5

Similar studies using panel data covered, for example, a period of 12 years (2008-2019 [75], 2004-2016 [84]). The literature does not indicate a minimum number of years for panel analyses to be valid [85, 86]. Also, Franses (2004) [87] explains how different the panels can be from one another. For example, in a condensed panel data set either the number of individuals is equal to 1 or the data have been averaged over the n individuals resulting in a single variable. As a rule, the more numerous the data in the panel, the more complex the model. A three-year panel is a simple one. Nevertheless, in the case of panel data analyses, the most relevant aspect is not the number of years, but the use of aggregated variables (instead of non-aggregated ones) [87].

As a result of pretesting, the fixed-effects model was found to be the most suitable for the input data. As known, the empirical model cannot be randomly selected.

The panel size, the time period and the methodology are the researcher’ choice, but they are also constraints deriving from the access to data. In the Conclusion section, discussing the limitations of the research, we emphasized these aspects. To extend the dataset and use a dynamic panel data model means to elaborate another study, a completely new one, and, moreover, without access to all the necessary data. But even so, according to Blackburne and Frank, Estimation of nonstationary heterogeneous panels, The Stata Journal, 2007, 7(2), 197-208, the PMG method could not be applied, because the number of the Black Sea riparian countries is only 6 and the available statistical data do not cover a long enough period of time.

The amount of GHG emissions per inhabitant depends on many factors. Energy production and processing and transport of inhabitants (not tourism) are the largest sources. Please add other explanatory variables. The equation suffers from an "omitted variable bias".

Analyzing the effects of tourism is difficult because the value of the indicators is not objective. We would welcome any suggestions regarding new variables to be considered. Variables for which data are available. Equations 1 and 2 in the Methodology section show the existence of errors caused by the influence of certain variables not included in the analysis for various reasons.

The approach to modelling the impact of tourism on emissions is wrong. Please read carefully Lenzen et al. (2018) which is the seminal reference.

We have read the paper by Lenzen et al. (2018). Their conclusion is similar to ours, in the sense that the effects of tourism on the carbon footprint are rather negative. In addition, their paper analyzes the period 2009-2013. In recent years, there has been much pressure, especially in the EU, on decarbonisation and the implementation of measures necessary to achieve this objective. Their work is obviously very valuable, but we are not sure how their study dealing with 160 countries for the period 2009-2013 (the empirical methodology is quite unclear to us) could guide us in approaching, for the period 2005-2020, 3 countries not considered by Lenzen et al.

We will definitely consider the second paper when applying PMG.

Please elaborate on the contribution. There are so many empirical studies investigating the relationship between total co2 emissions and tourism using aggregate country level data.

Reference

Lenzen, M., Sun, Y. Y., Faturay, F., Ting, Y. P., Geschke, A., & Malik, A. (2018). The carbon footprint of global tourism. Nature Climate Change, 8(6), 522-528.

Pesaran, M. H., Shin, Y., & Smith, R. P. (1999). Pooled mean group estimation of dynamic heterogeneous panels. Journal of the American statistical Association, 94(446), 621-634.

In the second version of the paper please find 7 paragraphs (colored in blue) that were added as answers to your suggestions/observations.

Changes have been made regarding the English language.

Yours faithfully,

Authors

Reviewer 3 Report

The manuscript presents interesting research results. However, there are a few issues/suggestions to consider:

-recommend to use capital letters for the abbreviations of the variables

-line 246: this statement requires some proof perhaps in parenthesis the test used and measurements that indicate the choice of the model.

- line 274: it is not mentioned what dummy variable was used.

- for the theoretical model it is always nice and useful for young researchers to find a book reference

-lines 282-299 – should be moved at the end of the literature review section as it is a justification of the approach taken and not results of data analysis.

- regressions results: 3-4 decimal points are sufficient (please check other scientific articles published in the journal)

- results section: tables do not require notes like “own calculation..” since it is expected and normal to find the authors own results in an original research article. Also, the data were already pointed in the previous section, thus it is not necessary to keep indicating the source of data

- results section: it is not very clear which are the steps taken for each hypothesis investigation, which actually are a bit overlapping. Thus, please revise. My suggestion is to first remind the hypothesis in words, mention what model was used and why and after present the results. This flow should be found for each of the 3 hypotheses.

- why is the methodology a limitation? What could have been don differently? What us it missing?

Author Response

Dear Sir / Madam,

We received your evaluation on the paper Longitudinal analysis of sustainable tourism potential of the Black Sea riparian states Bulgaria, Romania and Turkey. The suggestions you made helped us improve our work. Thank you for your time and effort!

Please find below the answers to your observations (colored in green, like in the paper).

The manuscript presents interesting research results. However, there are a few issues/suggestions to consider:

-recommend to use capital letters for the abbreviations of the variables

The abbreviations are now written in capital letters.

-line 246: this statement requires some proof perhaps in parenthesis the test used and measurements that indicate the choice of the model.

We believe you indicated lines 245-249 of the initial version of the paper ("Taking Turkey as a model for tourism development, Bulgaria and Romania should adopt appropriate tourism development strategies, attractive tourism packages, provide visitors with easy access to representative and quality goods, all this under sustainability conditions.")

We rephrased the first part:

Considering Turkey as a reference point for tourism development in the Black Sea riparian region, Bulgaria and Romania should adopt appropriate tourism development strategies, attractive tourism packages, provide visitors with easy access to specific quality goods, all this under sustainability conditions.

By " Considering Turkey … riparian region" we mean that Turkey is a model for tourism development in the region. This conclusion comes from statistical data and from literature findings.

- line 274: it is not mentioned what dummy variable was used.

Line 274 refers to error terms (uit = μi+ ηi + vit (2)) (not dummy variables). These terms are explained between lines 342 and 352 (in the new version of the paper). The analysis does not contain dummy variables because the model is one with fixed effects. We spoke about that between lines 291 and 293 in the initial version of the paper.

- for the theoretical model it is always nice and useful for young researchers to find a book reference

We added 4 references:

  1. Cang, S., & Seetaram, N. (2012). Time series analysis. In L. Dwyer, A. Gill, N. Seetaram (Eds.), Handbook of Research Methods in Tourism: Quantitative and Qualitative Approaches (pp. 47-70). Cheltenham: Edward Elgar Publishing.
  2. Veal, A. J. (2018). Research Methods for Leisure and Tourism. Harlow: Pearson Education. https://nibmehub.com/opac-service/pdf/read/Research%20Methods%20for%20Leisure%20and%20Tourism.pdf
  3. Franses, P. H. (2004). A Concise Introduction to Econometrics: An Intuitive Guide. Cambridge: Cambridge University Press.
  4. Adkins, L. C, & Hill, R. C. (2011). Using Stata for Principles of Econometrics. New York: John Wiley & Sons.

- regressions results: 3-4 decimal points are sufficient (please check other scientific articles published in the journal)

We reduced the decimal points to 4.

- results section: tables do not require notes like “own calculation” since it is expected and normal to find the authors own results in an original research article. Also, the data were already pointed in the previous section, thus it is not necessary to keep indicating the source of data

We removed this phrase from tables and graphs.

- results section: it is not very clear which are the steps taken for each hypothesis investigation, which actually are a bit overlapping. Thus, please revise. My suggestion is to first remind the hypothesis in words, mention what model was used and why and after present the results. This flow should be found for each of the 3 hypotheses.

We made these comments at the suggestion of Reviewer 1, but we placed them after the hypotheses were validated/invalidated (not before). We added explanations between lines 476-485, 504-508 and 692-696.

If you consider that the comments regarding the hypotheses should be placed at the beginning of the empirical approach (not at the end), we will make the change.

- why is the methodology a limitation? What could have been don differently? What us it missing?

We added a short explanation on the methodology limitations (lines 756-757). We consider that any method used in a scientific study is a limitation in itself. A different model, the inclusion or exclusion of some variables, a different period of time, all these represent limitations of any research endeavor.

We hope that we have managed to answer your requirements. We will try to further improve the text if necessary.

Yours faithfully,

Authors

Round 2

Reviewer 2 Report

Thank you for the revision. CO2 emissions are mainly caused by the burning of fossil fuels. Travel and tourism contribute only marginally to total CO2 emissions. The equation suffers from an omitted variables bias. The paper also has nothing to do with sustainable tourism. The specification contains many variables that are highly correlated. The revisions do not meet my expectations. Please see my previous comments.

Author Response

Thank you for the revision. CO2 emissions are mainly caused by the burning of fossil fuels. Travel and tourism contribute only marginally to total CO2 emissions. The equation suffers from an omitted variables bias. The paper also has nothing to do with sustainable tourism. The specification contains many variables that are highly correlated. The revisions do not meet my expectations. Please see my previous comments.

  • All literature shows that tourism sustainability is evaluated with the help of emissions (GHG or CO2). In addition, tourism has brought sustainability into the economic growth equation.
  • The dependent variable is Green House Gas Emissions not CO2. GHG are mesured in kt of CO2 equivalent but GHG are not CO2.
  • The method proposed by you does not fit to the data. We stick to this variant of the methodology. The data have an average degree of correlation (corr(u_i, Xb)  = 0.1798), and the methodology is mentioned at the limits of the research

Reviewer 3 Report

Dear authors, the review was made based on the manuscript uploaded by you on the platform. Thus, it is normal to read and refer to any suggestions according to that version.

Comments that still need to be clarified:

1) This statement requires some proof perhaps in parenthesis the test used and measurements that indicate the choice of the model: “After testing the Fixed Effects and the Random Effects regression models, we chose the first one” (line 331 in current version = line 246 in the first version) - recommend to add the tests perfomed (results are presented but in the methodology section should be added which ones were used)

2) Fixed effects models CAN BE also with dummy variables (I recommend to see theory for clarification for future work). Now in this manuscript, if the dummies were not used then why was presented the theory related to dummies, a model used in the initial version of the paper? Which one is the initial version? The manuscript should present only the methodology applied to the results presented in this manuscript. (The comment in the first review was based on: “ In equation (3), dij = 1 if i = j or dij = 0 in other way” (line 360 in current version = line 274 in the first version)). 

Author Response

Comments that still need to be clarified:

1) This statement requires some proof perhaps in parenthesis the test used and measurements that indicate the choice of the model: “After testing the Fixed Effects and the Random Effects regression models, we chose the first one” (line 331 in current version = line 246 in the first version) - recommend to add the tests performed (results are presented but in the methodology section should be added which ones were used)

  • In the paragraph below table 2, we specified that, after applying the Hausmann test, we choose Fixed Effects. With p value = 0.0000, we choose Fixed Effects. If p value would have been statistically insignificant, then we would have chosen the Random Effect model. We have added the results of the Pesaran`s test and Modified Wald test in Table 2. We briefly explained the results obtained and the fact that they, together, led us to choose FE Model.
  • We added some specifications regarding the Hausman, Pesaran and Wald tests in the Methodology section.

2) Fixed effects models CAN BE also with dummy variables (I recommend to see theory for clarification for future work). Now in this manuscript, if the dummies were not used then why was presented the theory related to dummies, a model used in the initial version of the paper? Which one is the initial version? The manuscript should present only the methodology applied to the results presented in this manuscript. (The comment in the first review was based on: “ In equation (3), dij = 1 if i = j or dij = 0 in other way” (line 360 in current version = line 274 in the first version)).

  • You are right. We took over the theoretical model with dummy variables, and later we specified that we do not include them in the analysis. We have deleted the respective details, and in the future we will be careful not to repeat this mistake.